# Frequency and Effectiveness of Empirical Anti-TNF Dose Intensification in Inflammatory Bowel Disease: Systematic Review with Meta-Analysis

**DOI:** 10.3390/jcm10102132

**Published:** 2021-05-14

**Authors:** Laura Guberna, Olga P. Nyssen, María Chaparro, Javier P. Gisbert

**Affiliations:** 1Gastroenterology Unit, Hospital Universitario de La Princesa, Instituto de Investigación Sanitaria Princesa (IIS-IP), Universidad Autónoma de Madrid (UAM), 28006 Madrid, Spain; lauraguberna@hotmail.com (L.G.); opn.aegredcap@aegastro.es (O.P.N.); mariachs2005@gmail.com (M.C.); 2Centro de Investigación Biomédica en Red de Enfermedades Hepáticas y Digestivas (CIBERehd), 28029 Madrid, Spain

**Keywords:** inflammatory bowel disease, Crohn’s disease, ulcerative colitis, anti-TNF-α, loss of response, dose intensification

## Abstract

Loss of response to antitumor necrosis factor (anti-TNF) therapies in inflammatory bowel disease occurs in a high proportion of patients. Our aim was to evaluate the loss of response to anti-TNF therapy, considered as the need for dose intensification (DI), DI effectiveness and the possible variables influencing its requirements. Bibliographical searches were performed. Selection: prospective and retrospective studies assessing DI in Crohn’s disease and ulcerative colitis patients treated for at least 12 weeks with an anti-TNF drug. Exclusion criteria: studies using anti-TNF as a prophylaxis for the postoperative recurrence in Crohn’s disease or those where DI was based on therapeutic drug monitoring. Data synthesis: effectiveness by intention-to-treat (random effects model). Data were stratified by medical condition (ulcerative colitis vs. Crohn’s disease), anti-TNF drug and follow-up. Results: One hundred and seventy-three studies (33,241 patients) were included. Overall rate of the DI requirement after 12 months was 28% (95% CI 24–32, *I2* = 96%, 41 studies) in naïve patients and 39% (95% CI 31–47, *I2* = 86%, 18 studies) in non-naïve patients. The DI requirement rate was higher both in those with prior anti-TNF exposure (*p* = 0.01) and with ulcerative colitis (*p* = 0.02). The DI requirement rate in naïve patients after 36 months was 35% (95% CI 28–43%; *I2* = 98%; 18 studies). The overall short-term response and remission rates of empirical DI in naïve patients were 63% (95% CI 48–78%; *I2* = 99%; 32 studies) and 48% (95% CI: 39–58%; *I2* = 92%; 25 studies), respectively. The loss of response to anti-TNF agents―and, consequently, DI―occurred frequently in inflammatory bowel disease (approximately in one-fourth at one year and in one-third at 3 years). Empirical DI was a relatively effective therapeutic option.

## 1. Introduction

Biologic therapies have become the mainstay of treatment in inflammatory bowel disease (IBD). Antibodies targeting tumor necrosis factor-alpha (anti-TNF) have become essential in the armamentarium for the treatment of both ulcerative colitis (UC) and Crohn’s disease (CD). TNF is a key proinflammatory cytokine that plays an important role in several autoimmune disorders, including IBD. Elevated stool and mucosal TNF concentrations in UC and CD patients have been shown to correlate with the disease activity [1]. Anti-TNF drugs operate via a multitude of mechanisms: they bind and clear soluble TNF but, also, cell-bound TNF, inducing cytotoxicity on immune cells, like T-cell apoptosis [2]. They are effective at inducing symptom relief, disease remission and mucosal healing and reducing the need for surgery and hospitalizations among patients with moderate-to-severe IBD. The current clinical guidelines recommend anti-TNF agents for patients who are refractory to other treatments [3,4,5,6].

However, a considerable proportion of these patients does not respond to induction therapy (primary nonresponse) or lose response over time (secondary nonresponse or loss of response, LOR). In patients who experience LOR to a particular anti-TNF agent, dose escalation or intensification (DI), either by increasing the dose or decreasing the dosing intervals, is commonly used as a rescue strategy to regain the therapeutic effect. Nevertheless, the exact incidence and chronology of this intensification, and its efficacy, are still not well-known.

The aim of this systematic review was to evaluate the incidence of LOR (defined as the need for DI) over time and DI efficacy in regaining both the response and remission in inflammatory bowel disease. The secondary objectives were to identify the possible variables (baseline medical condition, anti-TNF therapy and time of follow-up) influencing the DI requirement and its efficacy.

## 2. Materials and Methods

### 2.1. Literature Search and Study Selection

Bibliographic searches were performed in four electronic databases (Medline, Embase, Cochrane Library CENTRAL and CINAHL) from inception up to January 2020. The search strategy (with corresponding keywords in all fields) was: “(inflammatory bowel disease OR Crohn’s disease OR ulcerative colitis) AND (infliximab OR adalimumab OR certolizumab OR golimumab OR antiTNF OR anti-TNF) AND (intensification OR escalation OR optimization OR optimisation)”. Additional hand searches were performed by the cross-referencing of eligible studies in order to identify further relevant publications. Abstracts were screened to discard duplicates. When the literature search yielded two or more studies by the same author assessing the same populations, only the most recent one was chosen, irrespective of the time interval, as it was assumed the latter published would include the most comprehensive and complete data.

The process of study selection is depicted in a flow diagram following the PRISMA statement [7]. The present systematic review was registered in PROSPERO (CRD42017073757). The selection process, data extraction and analyses were performed by two authors (LG and OPN) independently. If discrepancies occurred, consensus was reached by a third reviewer (JPG). The corresponding authors of the studies without sufficient data were contacted for additional information.

### 2.2. Selection Criteria

Prospective and retrospective studies assessing the LOR to anti-TNF therapy, considered as the need for DI in patients with CD and UC treated for at least 12 weeks with an anti-TNF drug, were selected for inclusion. There were no language restrictions.

Articles in which an anti-TNF was used as the prophylaxis for postoperative recurrence in CD and those where DI was based during therapeutic drug monitoring were excluded. Systematic or narrative reviews, case studies and congress abstracts were excluded from this systematic review.

### 2.3. Data Extraction and Quality Assessment

A predefined, pre-piloted data extraction form was used to collect the data. The variables recorded were: year of publication; study design (prospective or retrospective); age of the study population (adults ≥ 18 years and children < 18 years); type of inflammatory bowel disease (UC or CD); therapeutic regimens (infliximab (IFX), adalimumab (ADA), certolizumab-pegol, and golimumab); previous anti-TNF treatments (naïve or non-naïve); length of follow-up in months; sample size; and outcome measures (DI requirement and DI efficacy).

The Cochrane risk of bias tool [8] was used to assess the quality of the randomized controlled trials, as they were considered the most reliable method of outcome assessment. The decision was reached post-hoc after performing an exploratory mapping review and confirming the wide range of observational studies in terms of the number and design available in the literature responding to our topic of interest.

### 2.4. Data Synthesis and Statistical Analysis

All analyses were preplanned a priori. The primary outcomes were the DI requirement measured as the number of patients receiving a DI out of the total of patients studied and DI efficacy in the short term as the number of patients responding out of the total of patients receiving a DI, expressed as the response rate with its standard error. These outcomes were thereafter combined using the inverse variance method, providing 95% confidence intervals (CIs). The statistical significance threshold was set at *p*-value < 0.05. A random effects model was used.

The study heterogeneity was analyzed using the *I2* statistic: according to the *I2* values, the heterogeneity was considered as: not important (*I2* < 40%), moderate (40–75%) and considerable (>75%). Such interpretations also adjusted for the magnitude of the effect and/or the strength of the evidence given (i.e., *p*-value < 0.1 of the χ^2^ test). Begg’s funnel plot [9] was used to estimate the possibility of publication bias.

Post-hoc sensitivity analyses were performed for each meta-analysis subgroup by excluding those studies that were identified as potentially introducing a critical risk of bias that could likely modify the outcome.

Data were analyzed using the Review Manager program (version 5.2).

## 3. Results

A total of 173 studies (including 33,241 patients) met the inclusion criteria and were finally included in the systematic review and meta-analysis (Figure 1).

The description of each included study is summarized in Table 1.

There were six randomized, placebo-controlled trials (RCTs) [10,11,12,13,14,15], 48 prospective open-label observational trials and 119 retrospective studies.

A total of 157 studies assessed the need for DI; the response rate was evaluated in 52 studies, and the remission rate was reported in 33 studies.

One hundred and one studies focused on naïve patients, and 29 evaluated non-naïve patients, while 50 studies included both naïve and non-naïve patients in their assessments. In six studies, prior anti-TNF exposure was not reported. One hundred and seven studies reported the data from IFX users and 92 from ADA users. Only five studies included patients receiving golimumab [16,17,18,19,20], and four studies evaluated patients receiving certolizumab [21,22,23,24]; thus, a meta-analysis was not performed.

### 3.1. Dose Intensification Requirements

#### 3.1.1. Twelve-Month Follow-Up

##### Naïve vs. Non-Naïve Patients

A total of 68 studies with a median follow-up of 12 months were analyzed.

In naïve patients, the DI rates ranged from 2% (100) to 80% (165), with an overall pooled rate of 28% (95% CI 24-32, *I2* = 96%, 41 studies) (Figure 2).

In non-naïve patients, the DI rate ranged from 7% (26) to 81% (111), with an overall pooled rate of 39% (95% CI 31-47, *I2* = 86%, 18 studies) (Figure 2).

The DI requirement after the 12-month follow-up was statistically higher in non-naïve than in naïve patients (test for subgroup differences: χ^2^ = 6.13, *p* = 0.01, *I2* = 83.7%).

##### Anti-TNF Use by Medical Condition in Naïve Patients

The DI requirement rate after the 12-month follow-up with all the anti-TNF agent data was statistically higher in UC than in CD patients (test for subgroup differences: χ^2^ = 5.29, *p* = 0.02, *I2* = 81.1%). No other subgroup differences were reported by the medical condition or anti-TNF used (Table 2).

#### 3.1.2. Thirty-Six Month Follow-Up

A total of 25 studies with a median follow-up of 36 months were analyzed. There was only one study reporting the DI rate in non-naïve patients, and therefore, no subgroup analysis was performed.

The DI rates in naïve patients ranged from 0% (113) to 70% (183), with an overall rate of 35% (95% CI 28–43%, *I2* = 98%, 18 studies) (Figure 3).

##### Anti-TNF Use by Medical Condition in Naïve Patients

No statistical differences (*p* > 0.05) in the medical conditions or the anti-TNF drug used were found between the subgroups (Table 3).

#### 3.1.3. Short-Term Follow up

A total of 17 studies with a median of three to nine months of follow-up were included. The DI rates in naïve patients ranged from 14% (130) to 71% (79) with an overall pooled rate of 29% (95% CI 31–37, *I2* = 96%, five studies).

A subgroup analysis evaluating the follow-up time (short-term vs. 12 months vs. 36 months) showed no statistical differences (*p* > 0.05) in terms of the DI requirements in naïve patients.

### 3.2. Dose Intensification Efficacy

#### 3.2.1. Response Rate

The response rates ranged from 0% (147) to 96% (48) in naïve patients and from 41% (60) to 75% (181) in non-naïve patients.

The overall rate of the short-term response to the empirical DI was 63% (95% CI: 48–78%, *I2* = 99%, 32 studies) and 58% (95% CI: 47–70%, *I2* = 68%, nine studies) in the naïve and non-naïve patients, respectively (Figure 4). No statistical differences were found between the groups (*p* > 0.05).

No statistical differences were found when comparing CD vs. UC patients or the anti-TNF drugs used (Table 4). Neither were found (*p* > 0.05) between different intensification regimens (i.e., intensification of dosing vs. reduction of the interval of administration).

#### 3.2.2. Remission Rate

The remission rates ranged from 17% (168) to 94% (183) in naïve patients and from 17% (60) to 85% (124) in non-naïve patients. The overall remission rate to empirical DI was 48% (95% CI: 39–58%, *I2* = 92%, 25 studies) and 44% (95% CI: 17–71%, *I2* = 95%, six studies) in naïve and non-naïve patients, respectively (Figure 5). No significant differences were found between the subgroups (*p* > 0.05).

No statistical differences were found when comparing CD vs. UC patients or the anti-TNF drugs used (Table 5). Neither were found between the different intensification regimens.

### 3.3. Pediatric Population

A total of 24 studies reported data on children (<18 years) (Table 1). When compared to the adult population, no statistical differences were found in terms of the DI required or its efficacy. The random-effects pooled DI rate in naïve patients after a 12-month follow-up was 29% (95% CI 21–37%, *I2* = 81%, *n* = 9).

### 3.4. Randomized Controlled Trials

A total of five randomized controlled trials (Table 1) assessed the DI requirements after a 12-month follow-up in naïve patients. The random-effects pooled DI rate was 29% (95% CI 18–41%. *I*2 = 88%, five studies). No statistical differences were found when this subgroup was compared to the group of observational studies.

### 3.5. Sensitivity Analyses and Risk of Bias

We further investigated potential sources of heterogeneity by excluding studies that included extreme or diverging values in certain subgroups, such as the DI requirements after 12 months [34,100,123,127,147,165] and 36 months [85,113] of follow-up or the response [147] and remission [61,145,168,183] rates. The effects of including different follow-up periods in the same subgroup [34,147,149] or the use of different induction dosing regimens [13,126,138] were also explored. In all cases, the results were stable, with no significant variations after the sensitivity analysis, although the heterogeneity remained considerable.

Among the six RCTs evaluated for a potential risk of bias, five had a low risk of bias for randomization, and four of them reported on the implementation of the random allocation sequence preserving concealment. Four studies also reported the adequate blinding of participants and personnel. Three studies showed low risks of attrition bias; in two of them, the number of excluded patients was not specified, and in the remaining one, there was a difference in the proportion of the outcome data. Finally, none of the studies was considered to show reporting biases. In conclusion, for most of the RCT items assessed, there was a low potential risk of bias detected.

## 4. Discussion

A LOR to the anti-TNF agents represents a therapeutic challenge to gastroenterologists, as these drugs are usually indicated in severe forms of the disease, and the remaining treatment options in such situations are limited. However, there is no unanimous definition of LOR in the literature [185,186]; it has been defined as an increase in clinical activity (which can be assessed by numerous activity indices) or, alternatively, as the need to modify or discontinue the current treatment. Thus, several authors have proposed that the DI requirement, which has been shown to recapture the response in multiple studies [187], would be a more objective and reliable measure [188] and, therefore, a useful surrogate for the LOR. Several reviews have previously assessed the incidence of a LOR, mainly in CD [185,186,187,188,189,190,191]. When compared to previous reviews, our study includes a considerably higher number of studies, up to January 2020, assessing both UC and CD patients and, therefore, conferring more robustness and reliability to our work.

### 4.1. Prior Anti-TNF Exposure

Several studies have estimated that approximately one-third of inflammatory bowel disease patients experience LOR and require DI, and that occurs more frequently in patients with prior anti-TNF exposure [188,189,190,191].

In our study, the overall rate of the DI requirements at a one-year follow-up was 28% in naïve and 39% in non-naïve patients, respectively. This shows no relevant differences with the previous data and constitutes one main finding of our study: dose escalation was needed more often in patients with prior anti-TNF use. In fact, the vast majority of the included studies evaluating both naïve and non-naïve patients showed a greater incidence in the loss of response in those non-naïve [30,34,35,60,81,84,111,121,132,161,163,168,192,193].

### 4.2. Time of Follow-Up

Additionally, the time course of LOR remains poorly understood. The median time from the first anti-TNF exposure to the need for a DI varied widely among the studies, from 2.7 to 18 months. However, there is increasing evidence showing that such events occur mostly within the first year of anti-TNF therapy [186].

In our study, no differences were found in the rate of DI for the short term, 12 and 36 months of follow-up, supporting the fact that the LOR and consequent DI occur mainly during the first year of treatment.

### 4.3. Medical Baseline Condition

Another relevant finding in our study was that a DI was required more frequently in UC than in CD patients. Previous data indicated that some patients with active UC have a higher inflammatory burden and accelerated anti-TNF clearance [194,195,196]; therefore, they could require a higher drug exposure to achieve a response to TNF antagonists. This could be the rational explanation UC patients need for an earlier and more frequent DI than CD patients [110,120,167]. However, there is also evidence not supporting these results [174]. Further research should be conducted, as no randomized trials have focused on this subgroup of patients; they seem to have the highest DI rate and could benefit the most from alternative treatment strategies.

### 4.4. Anti-TNF Agent

The comparison between the IFX and ADA DI rates is also a matter of interest. Immunogenicity is believed to be a common cause of LOR due to the formation of antidrug antibodies. Some authors have argued that the chimeric nature of IFX, as opposed to the fully humanized ADA, could render the former more prone to generate an antibody response. However, in our study, we did not find significant differences in the DI rate between IFX and ADA patients, as in previous comparative reports [115].

### 4.5. Dose Intensification Efficacy

Several clinical trials and open-label cohorts included in a previous review reported DI to restore the response in 50–70% of patients [186]. Billioud et al. also found that DI restored the response in 71% and remission in 40% of the patients [189].

In our study, the response and remission rates to empirical DI in naïve patients were 63% and 48%, respectively. Although no significant differences were reported between the naïve and non-naïve patients, either in the response or remission rates, a trend towards a reduced DI efficacy in the patients with prior anti-TNF exposure was shown.

Our findings support that using all the available treatment options with the first anti-TNF agent through DI (even if it is not based on therapeutic drug monitoring) should be considered before switching to another anti-TNF agent or to another therapeutic target. Nevertheless, it should be noted that almost all studies do assess the DI efficacy in the short term; additional research regarding the long-term response and remission rates after DI should be performed.

### 4.6. Limitations

Our study had some limitations. First of all, the DI can result in an equivocal interpretation of the LOR if it is done without accurately confirming the disease activity. In addition, there were also some possible predictors for the LOR or DI that were not evaluated in our study, such as the concomitant use of immunomodulators. However, recent guidelines (three) have suggested monotherapy with anti-TNF in patients with long-term remission rather than the use of a combination therapy. Finally, we excluded studies in which the DI was made based on therapeutic drug monitoring, with the aim to assess the effectiveness of empirical DI. In this respect, the current guidelines (three) do not recommend either proactive or reactive therapeutic drug monitoring as a standard clinical practice due to insufficient evidence. Finally, we did not perform a quality assessment of all the included studies given the high heterogeneity of the observational studies encountered in terms of the design and number. It was decided to perform a risk of bias assessment exclusively in RCTs, which represented no more than 1.5% of the total of patients included in our systematic review but, including 512 patients, was a sufficient sample size to drawn robust conclusions. In terms of quality, most studies showed a low risk of bias for the majority of the items assessed, highlighting both an adequate random sequence generation and allocation concealment, as well as blinding: items that were usually preserved. Additionally, a subgroup analysis was performed to control for heterogeneity in terms of study design, and no significant differences in the DI requirement between the RCTs and observational studies were reported.

## 5. Conclusions

A LOR to anti-TNF agents―and, consequently, DI―occurs frequently in inflammatory bowel disease, with an overall rate of DI requirement of approximately one-fourth at one year and one-third at three years. DI is required more frequently in patients with prior exposure to anti-TNF agents and in UC patients. Empirical DI is a relatively effective therapeutic option, achieving a response in two-thirds and remission in one-half of those patients naïve to anti-TNF treatment.

## Figures and Tables

**Figure 1 jcm-10-02132-f001:**
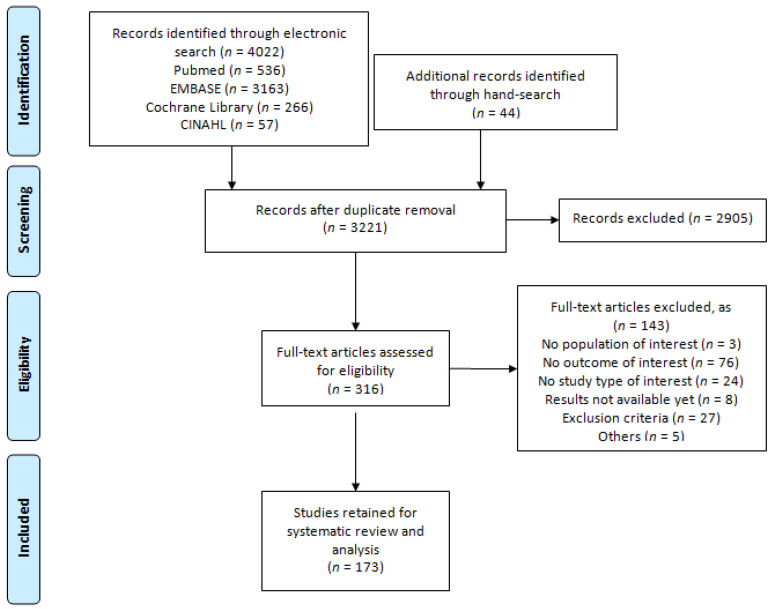
PRISMA flowchart of the screening and selection.

**Figure 2 jcm-10-02132-f002:**
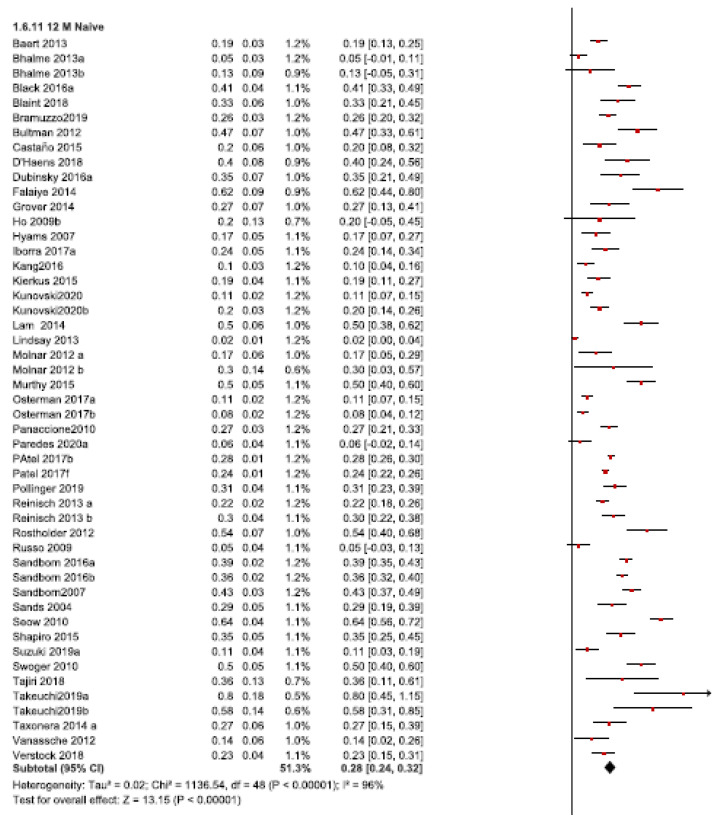
Dose intensification requirements after the 12-month follow-up in anti-TNF naïve and non-naïve patients.

**Figure 3 jcm-10-02132-f003:**
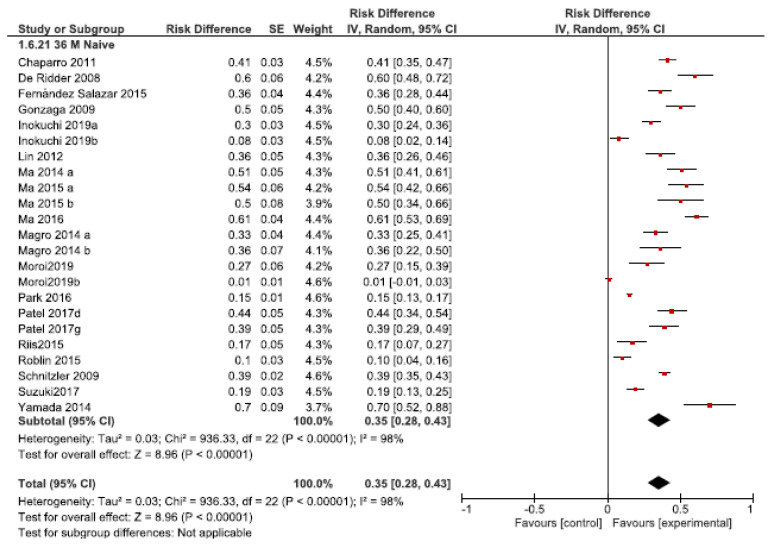
Dose intensification requirements after the 36-month follow-up in anti-TNF naïve patients.

**Figure 4 jcm-10-02132-f004:**
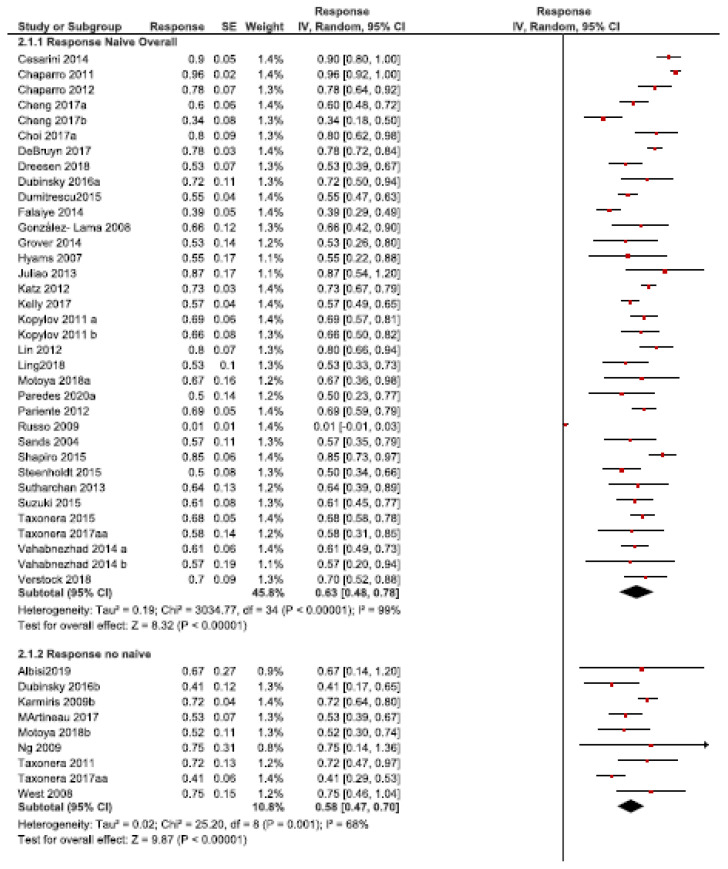
Response rate after the empirical dose intensification in anti-TNF naïve vs. non-naïve patients.

**Figure 5 jcm-10-02132-f005:**
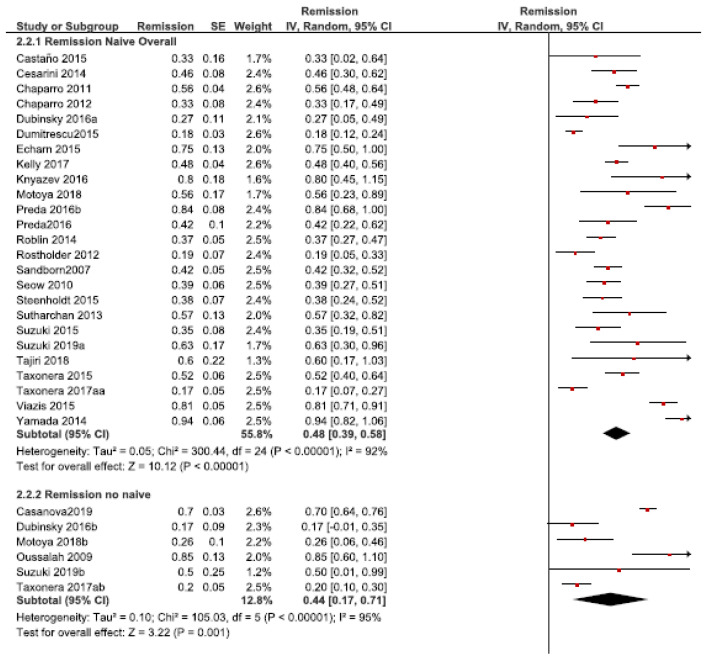
Remission rates after the empirical dose intensification in anti-TNF naïve vs. non-naïve patients.

**Table 1 jcm-10-02132-t001:** Studies included in the meta-analysis.

	Author and Year	Study Design	Population	Medical Condition	Anti-TNF	Prior Anti-TNF	FOLLOW up (Months)	*n*	N	DI Rate (%)	Intensification Regimen	Response/Remission	n’	N’	DI Efficacy (%)
1	Afif 2009 [25]	P	A	UC	ADA	Naïve and non-naïve	6	7	20	35					
2	Albisi 2019 [26]	R	C	CD	ADA	Non-naïve	12	3	44	7	ID	Response	2	3	67
3	Armuzzi 2013 [27]	R	A	UC	ADA	Naïve and non-naïve	12	31	88	35					
4	Assa 2013 [28]	R	C	UC+CD	IFX+ADA	-	20	10	102	10					
5	Baert 2014 [29]	R	A	UC	ADA	Non-naïve	12	22	73	30					
6	Baert 2013 [30]	R	A	CD	ADA	Naïve and non-naïve	14	208	605	34	RI	Response	139	208	67
				CD	ADA	Naïve	14	40	208	19					
				CD	ADA	Non-naïve	14	164	365	45					
7	Baki 2015 [31]	R	A	UC	IFX	Naïve and non-naïve	5	26	54	48					
				UC	ADA	Naïve and non-naïve	4	17	37	46					
8	Balint 2018 [32]	P	A	UC	IFX	Naïve	12	20	61	33					
9	Balint 2016 [33]	P	A+C	UC	ADA	Naïve and non-naïve	12	13	73	18					
10	Bhalme 2013 [34]	R	A	CD	IFX	Naïve	13	4	76	5					
					ADA	Naïve	11	2	15	13					
					ADA	Non-naïve	11	9	39	23					
					ADA	Naïve and non-naïve	11	11	54	20					
11	Black 2016 [35]	R	A	UC	ADA	Naïve	12	66	155	43					
					ADA	Non-naïve	12	17	36	47					
12	Bor 2017 [36]	R	A	CD	IFX	Naïve and non-naïve	-	14	48	29	ID	Remission	3	14	21
13	Bortlik 2013 [37]	R	A	CD	IFX	Naïve and non-naïve	24	6	84	7					
14	Bossuyt 2019 [38]	P	A	UC	GOL	Naïve and non-naïve	6	8	91	9					
15	Bouguen 2015 [39]	P	A	CD	ADA	Naïve and non-naïve					-	Response	23	42	55
												Remission	14	42	33
16	Bramuzzo 2019 [40]	R	C	UC+CD	IFX	Naïve	12	44	172	26					
17	Brandes 2019 [41]	R	A	UC+CD	ADA	Naïve and non-naïve	12	76	502	15					
18	Bultman 2012 [42]	P	A	CD	ADA	Naïve	12	23	49	47	-	Response	20	46	43
				CD	ADA	Non-naïve	12	23	73	31.5					
19	Cameron 2015 [43]	R	C	UC+CD	IFX	Naïve	23	23	72	32					
				UC+CD	ADA	Naïve and non-naïve	14	19	29	66					
20	Casanova 2019 [21]	R	A	UC+CD	IFX+ADA+CZP	Non-naïve	18	230	1122	20.5	RI or ID	Remission	161	230	42
21	Casellas 2015 [44]	P	A	CD	ADA	Naïve and non-naïve	36	3	28	11					
22	Castaño 2015 [45]	R	A	CD	ADA	Naïve	12	9	46	20	RI	Remission	3	9	33
23	Caviglia 2007 [46]	R	A	UC	IFX	-	24	0	10	0					
				CD	IFX	-	24	3	40	7.5					
24	Cesarini 2014 [47]	R	A	UC	IFX	Naïve	24				RI or ID	Response	37	41	90
												Remission	19	41	46
											RI	Response	24	26	92
												Remission	9	26	35
											ID	Response	13	15	87
												Remission	10	15	67
25	Chaparro 2011 [48]	R	A	CD	IFX	Naïve	41	127	309		RI + ID	Response	122	127	96
												Remission	71	127	56
26	Chaparro 2012 [49]	R	A	CD	IFX	Naïve	22	33	197	17	-	Response	26	33	79
												Remission	11	33	33
27	Cheng, 2017 [50]	R	C	UC	IFX	Naïve	24	60	113	53	RI or ID	Response	36	60	60
				CD	IFX	Naïve	24	19	35	54	RI or ID	Response	12	35	34
28	Choi 2014 [51]	R	A	CD	ADA	Naïve	18	5	36	14					
					IFX	Naïve	18	0	36	0					
29	Choi 2017 [52]	R	C	CD	IFX	Naïve	16	14	29	48	RI or ID	Response	17	21	80
				UC	IFX	Naïve	16	7	10	70					
30	Church 2014 [53]	R	C	CD	IFX	Naïve	21	79	157	50					
31	Clark 2019 [54]	R	A	CD	IFX	Non-naïve	24	10	17	59					
32	Cohen 2012 [55]	R	A	CD	ADA	Naïve and non-naïve	55	31	75	41					
33	Cordero 2011 [56]	P	A	CD	ADA	Non-naïve	12	18	25	72					
34	DeRidder 2008 [57]	R	C	CD	IFX	Naïve	41	40	66	61					
35	DeBruyn 2017 [58]	R	C	CD	IFX	Naïve	19	102	178	57					
36	D’Haens 2018 [10]	P	A	CD	IFX	Naïve	12	16	40	40					
37	Dignass 2019 [17]	R	A	UC	IFX	Naïve	24	75	114	66					
				UC	ADA	Naïve	24	49	125	39					
				UC	GOL	Naïve	24	27	47	57					
38	Dreesen 2018 [59]	R	A	CD	IFX	Naïve					RI, ID, RI+ID	Response	65	103	63
											ID	Response	24	45	53
											RI	Response	33	45	73
											RI + ID	Response	8	13	61
39	Dubinsky 2016 [60]	P	C	CD	ADA	Naïve and non-naïve	12	35	93	38	RI	Response	20	35	57
											RI	Remission	11	35	31
						Naïve	12	18	51	35	RI	Response	13	18	72
											RI	Remission	5	18	28
						Non-naïve	12	17	42	40	RI	Response	7	17	41
											RI	Remission	3	17	18
40	Dumitrescu 2015 [61]	R	A	UC	IFX	Naïve					RI or ID	Response	87	157	55
												Remission	28	157	18
41	Dupont 2016 [62]	R	C	CD	IFX	Naïve	-	65	187	35					
42	Duveau 2016 [63]	R	A	CD	ADA	Naïve and non-naïve	-	124	430	29	RI or ID	Response	99	124	80
43	Echarri 2015 [64]	P	A	CD	ADA	Naïve	24	12	68	18	RI	Remission	9	12	75
44	Falaiye 2014 [65]	R	A	UC+CD	IFX	Naïve	12	18	29	62	RI or ID	Response	7	18	39
45	Fernandes 2019 [66]	R	A	UC+CD	IFX	Naïve and non-naïve	12	25	149	17					
				UC+CD	IFX	Naïve and non-naïve	24	38	149	25.5					
46	Fernández-Salazar 2015 [67]	R	A	UC	IFX	Naïve	38	53	144	37					
47	Fiorino 2017 [68]	P	A+C	UC+CD	IFX	Naïve and non-naïve	3	74	399	16					
48	Fortea-Ormaechea 2011 [69]	R	A	CD	ADA	Naïve and non-naïve	9	57	174	33					
49	Frederiksen 2014 [70]	R	A	UC+CD	ADA	No naïve	9	21	57	37					
50	García bosch 2013 [71]	R	A	UC	ADA	Naïve and non-naïve	12	18	48	37.5	-	Response	15	18	83
											-	Remission	8	18	44
51	Ghaly 2015 [72]	R	A	CD	IFX+ADA	Naïve and non-naïve		40	73		-	Response	40	73	55
52	Gofin 2019 [73]	R	C	CD	IFX+ADA	Naïve	19	18	98	18					
53	Gonczi 2017 [74]	P	A	UC+CD	ADA	Naïve and non-naïve	12	22	112	20					
							24	33	112	29					
54	Gonzaga 2009 [75]	R	A	CD	IFX	Naïve	49	56	111	50					
55	González Lama 2008 [76]	R	A	CD	IFX	Naïve	28	15	114	13	RI or ID	Response	10	15	67
56	Grover 2014 [77]	R	C	CD	IFX	Naïve	12	13	47	28	-	Response	7	13	54
57	Guerbau 2017 [78]	P	A	CD	IFX	Naïve and non-naïve	12	43	140	30					
58	Guidi 2018 [79]	P	A	UC+CD	IFX	Naïve	3	37	52	71					
59	Ho 2008 [80]	R	A	CD	ADA	Non-naïve	12	13	22	59					
60	Ho 2009 [81]	R	A+C	CD	ADA	Naïve and non-naïve	6	24	98	24					
				CD	ADA	Naïve	12	2	10	20					
				CD	ADA	Non-naïve	12	28	88	32					
				CD	ADA	Naïve and non-naïve	24	54	98	55					
61	Hussey 2016 [82]	R	A	UC	ADA	Naïve and non-naïve	19	13	55	24					
62	Hyams 2010 [83]	P	C	UC	IFX	Naïve	30	11	34	33					
63	Hyams 2007 [11]	P	C	CD	IFX	Naïve	12	9	52	17	ID	Response	5	9	56
64	Iborra 2017 [84]	R	A	UC	ADA	Naïve and non-naïve	12	93	263	35					
					ADA	Naïve	12	21	87	24					
					ADA	Non-naïve	12	72	176	41					
65	Inokuchi 2019 [85]	R	A	CD	IFX	Naïve	83	54	183	29.5					
				CD	ADA	Naïve	43	6	80	7.5					
66	Juillerat 2015 [86]	R	A	CD	IFX	Naïve and non-naïve	-	77	250	31					
67	Juliao 2013 [87]	R	A	UC	IFX	Naïve	27	4	28	14	RI	Response	4	4	100
68	Kang 2016 [88]	P	C	CD	IFX	Naïve	12	7	72	10					
69	Karmiris 2009 [89]	P	A	CD	ADA	Non-naïve	20	102	156	65	RI	Response	73	102	72
70	Katz 2012 [90]	R	A	CD	IFX	Naïve	-				RI or ID	Response	123	168	73
											RI	Response	37	56	66
											ID	Response	86	112	77
71	Kelly 2017 [91]	R	A	UC+CD	IFX	Naïve					RI or ID	Response	82	143	57
												Remission	69	143	48
72	Kierkus 2015 [12]	P	C	CD	IFX	Naïve	12	16	84	19					
73	Kiss 2011 [92]	R	A	CD	ADA	Naïve and non-naïve	12	33	201	16					
74	Knyazev 2018 [22]	P	A	CD	CRP	Naïve and non-naïve	24	3	39	8					
75	Knyazev 2016 [93]	R	A	UC	IFX	Naïve	-	5	45	11	-	Remission	4	5	80
76	Knyazev 2017 [94]	P	A	CD	ADA	Naïve and non-naïve	28	6	70	9					
77	Kopylov 2011 [95]	R	A	CD	IFX	Naïve					RI	Response	38	55	70
				CD	IFX	Naïve					ID	Response	26	39	67
78	Kunovski 2020 [96]	R	A	UC	IFX	Naïve	12	43	396	11					
					ADA	Naïve	12	34	172	20					
79	Lam 2014 [97]	R	A	CD	IFX	Naïve	12	34	68	50					
80	Lees 2009 [98]	R	A+C	UC+CD	ADA	Non-naïve	12	16	30	53					
81	Lin 2012 [99]	R	A	CD	IFX	Naïve	60	34	94	36	RI or ID	Response	24	30	80
82	Lindsay 2013 [100]	R	A+C	CD	IFX	Naïve	12	9	380	2					
					IFX	Naïve	24	19	380	5					
83	Lindsay 2017 [101]	R	A	UC	IFX+ADA		24	139	538	26					
				CD	IFX+ADA	Naïve	24	126	657	19					
84	Ling 2018 [102]	R	C	CD	IFX	Naïve	24	26	43	60	RI or ID	Response	14	26	54
85	Llaó 2016 [103]	P	A	UC	IFX	-	18	8	15	53					
86	Lofberg 2012 [104]	P	A	CD	ADA	Naïve and non-naïve	5	131	945	14	RI	Remission	46	131	35
87	Lopez Palacios 2008 [105]	R	A	CD	ADA	Non-naïve	24	6	22	27	RI	Response	4	6	66
88	Ma 2015 [106]	R	A	UC	IFX	Naïve	158	36	66	54					
				UC	ADA	Naïve	139	18	36	50					
89	Ma 2014 [107]	R	A	CD	IFX	Naïve	40	60	117	51					
				CD	ADA	Naïve	28	23	38	61					
				CD	ADA	Non-naïve	28	41	63	65					
90	Ma 2016 [108]	R	A	CD	IFX+ADA	Naïve	38	116	190	61					
91	Ma 2014 (bis) [109]	R	A	CD	ADA	Naïve and non-naïve	-				-	Response	74	92	80
92	Magro 2014 [110]	R	A	CD	IFX	Naïve	84	55	163	34					
				UC	IFX	Naïve	84	19	52	37					
93	Martineau 2017 [19]	R	A	CD	GOL	Non-naïve	18	51	115	44	-	Response	27	51	53
94	Merras 2016 [20]	P	C	CD	GOL	Non-naïve	*	1	6	17					
95	Molnar 2012 [111]	R	A	CD	IFX	Naïve	12	3	35	9					
				CD	ADA	Naïve	12	3	10	30					
				CD	ADA	Non-naïve	12	13	16	81					
96	Moon 2015 [23]	R	A	CD	CZP	Naïve and non-naïve	26	43	358	12					
97	Motoya 2018 [112]	P	A+C	CD	ADA	Naïve and non-naïve					RI	Response	16	28	57
				CD	ADA	Naïve and non-naïve					RI	Remission	10	28	35
					ADA	Naïve					RI	Response	6	9	67
					ADA	Naïve					RI	Remission	5	9	56
					ADA	Non-naïve					RI	Response	10	19	53
					ADA	Non-naïve					RI	Remission	5	19	26
98	Moroi 2019 [113]	R	A	CD	IFX	Naïve	36	17	62	27					
					ADA	Naïve	36	0	7	0					
99	Murthy 2015 [114]	R	A	UC	IFX	Naïve	12	59	116	51					
100	Narula 2016 [115]	P	A	CD	IFX	Naïve	24	35	251	14					
				CD	ADA	Naïve	24	9	111	8					
101	Nedelkopoulou 2018 [116]	R	C	UC	IFX	Naïve	20	2	10	20					
102	Ng 2009 [117]	P	A	CD	ADA	Non-naïve	12	2	7	29	RI	Response	2	2	100
103	Nichita 2010 [118]	R	A	CD	ADA	Naïve and non-naïve	12	13	55	24	RI or ID	Response	8	13	62
												Remission	6	13	46
104	Nuti 2014 [119]	R	C	CD	IFX+ADA	Naïve and non-naïve	36	27	78	35					
105	O’Donnell 2015 [120]	R	A+C	CD	IFX	Naïve	36	133	287	46					
				UC	IFX	Naïve	36	84	125	67					
106	Olivares 2019 [121]	P	A	UC+CD	ADA	Naïve	18.	15	33	45					
				UC+CD	ADA	Non-naïve	18.	37	53	70					
				UC	ADA	Naïve and non-naïve	6.	7	43	16					
				CD	ADA	Naïve and non-naïve	6.	21	43	49					
				UC	ADA	Naïve and non-naïve	18.	24	43	56					
				CD	ADA	Naïve and non-naïve	20.	28	43	65					
107	Orlando 2012 [122]	P	A	CD	ADA	Naïve and non-naïve	14.	15	110	14					
108	Osterman 2017 [123]	R	A	CD	IFX	Naïve	12	42	381	11					
					ADA	Naive	12	16	196	8					
109	Oussalah 2009 [124]	R	A	CD	ADA	Non-naïve	36	7	53	13	RI	Remission	6	7	86
110	Oussalah 2010 [125]	R	A	UC	IFX	Naïve	18	36	80	45					
111	Panaccione 2010 [126]	P	A	CD	ADA	Naïve	12	71	260	27					
							24	105	260	40					
112	Paredes 2020 [127]	P	A	UC+CD	IFX	Naïve	12	2	31	6					
				UC+CD	IFX	Naïve	24	12	31	39	-	Response	6	12	50
				UC	IFX	Naïve	24	3	31	10					
				CD	IFX	Naïve	24	9	31	29					
113	Pariente 2012 [128]	R	A	UC+CD	IFX	Naïve					RI or ID	Response	27	39	69
114	Park 2016 [129]	R	A	CD	IFX	Naïve	36	86	582	15					
115	Patel 2017 [130]	R	A	CD	IFX+ADA+ CZP+GOL	Naïve	6	640	4569	14					
						Naïve	12	1097	4569	24					
						Naïve	24	1553	4569	34					
						Naïve	36	1782	4569	39					
				UC	IFX+ADA+ CZP+GOL	Naïve	6	272	1699	16					
						Naïve	12	475	1699	28					
						Naïve	24	680	1699	40					
						Naïve	36	748	1699	44					
116	Paul 2013 [131]	P	A	UC+CD	IFX	Naïve and non-naïve					ID	Remission	30	52	58
117	Peters 2014 [132]	R	A	CD	ADA	Naïve	24	45	167	27					
				CD	ADA	Non-naïve	24	135	271	50					
118	Peyrin 2007 [133]	P	A	CD	ADA	Non-naïve	12	6	24	25					
119	Pollinger 2019 [134]	R	A	UC	ADA	Naïve	12	48	154	31					
120	Preda 2016 [135]	R	A	CD	IFX	Naïve	36	26	129	20	-	Remission	11	26	42
				CD	ADA	Naïve	20	19	136	14	-	Remission	16	19	84
121	Qazi 2016 [136]	P	A	UC+CD	IFX	Naïve	24	10	75	13					
122	Regueiro 2007 [137]	R	A	CD	IFX	Naïve and non-naïve	30	54	108	50	RI or ID	Response	41	54	76
123	Reinisch 2013 [138]	P	A	UC	ADA	Naïve	12	110	445	25					
124	Renna 2016 [139]	P	A	UC	ADA	Non-naïve	< 6	1	16	6					
125	Renna 2018 [140]	R	A	UC	ADA	Naïve and non-naïve	10	50	118	42	RI	Response	23	50	46
126	Riis 2012 [141]	R	A	CD	IFX	Naïve	59	10	58	17					
				CD	ADA	Naïve	36	1	19	5					
127	Roblin 2014 [142]	P	A	UC+CD	ADA	Naïve					RI	Remission	30	82	36
128	Roblin, 2016 [143]	P	A	CD	IFX	Naïve and non-naïve	20	30	119	25					
129	Roblin 2015 [144]	P	A	UC+CD	IFX	Naïve	20	10	93	11					
130	Rostholder 2012 [145]	R	A	UC	IFX	Naïve	12	27	50	54	RI or ID	Remission	5	27	19
131	Rubin 2012 [146]	R	A	CD	ANTI TNF	-	24	531	1398	38					
132	Russo 2009 [147]	R	A	UC	IFX	Naïve	15	2	38	5	RI or ID	Response	0	2	0
133	Rutka 2016 [148]			UC	ADA	Naïve and non-naïve	12	13	73	18					
134	Sandborn 2007 [13]	P	A	CD	ADA	Naïve	12	89	204	44	-	Remission	37	89	42
135	Sandborn 2016 [149]	R	A	UC	IFX	Naïve	11	166	424	39					
				UC	ADA	Naïve	11	138	380	36					
136	Sands 2004 [14]	P	A	CD	IFX	Naïve	12	28	96	29	RI	Response	12	21	57
137	Sartini 2018 [150]	R	A	UC	ADA	Naïve and non-naïve	24	17	32	53					
				CD	ADA	Naïve and non-naïve	24	58	149	39					
138	Sazuka 2012 [151]	R	A	CD	IFX	Naïve	21	30	74	40					
139	Schnitzler 2009 [152]	P	A	CD	IFX	Naïve	55	218	547	40					
140	Seo 2017 [153]	R	A	CD	ADA	Naïve and non-naïve	17	45	254	18					
141	Seow 2010 [154]	P	A	UC	IFX	Naïve	14	74	115	64	RI or ID	Remission	29	74	39
142	Shapiro 2015 [155]	R	C	UC+CD	IFX	Naïve	12	35	87	40	RI or ID	Response	30	35	86
143	Sierra 2016 [156]	R	A	CU	ADA	Naïve and non-naïve	12	16	37	43					
144	Sprakes 2012 [157]	P	A	CD	IFX	Naïve	24	18	173	10					
145	Srinivasan 2018 [158]	R	A	CD	IFX+ADA	Naïve and non-naïve	12	55	423	13					
146	Stein 2014 [24]	R	A	CD	CZP	Naïve and non-naïve	124	10	87	11					
147	Steendholt 2015 [15]	P	A	CD	IFX	Naïve					RI	Response	19	36	53
148	Sutharsan 2013 [159]	P	A	CD	ADA	Naïve					RI	Response	9	14	64
				CD	ADA	Naïve						Remission	8	14	57
159	Suzuki 2015 [160]	P		CD	IFX	Naïve					ID	Response	23	39	59
				CD	IFX	Naïve						Remission	13	39	36
150	Suzuki 2019 [161]	R	A	CD	ADA	Naïve and non-naïve	12	14	95	15	ID	Remission	8	12	67
				CD	ADA	Naïve	12	9	78	12		Remission	5	8	62.5
				CD	ADA	Non-naïve	12	5	17	29		Remission	2	4	50
151	Suzuki 2017 [162]	P	A	UC	ADA	Naïve	36	36	190	19					
152	Swoger 2010 [163]	R	A	CD	ADA	Naïve	12	59	118	50					
153	Tajiri 2018 [164]	P	C	CD	IFX	Naïve	12	5	14	36	ID	Remission	3	5	60
154	Takeuchi 2019 [165]	R	C	UC+CD	IFX	Naïve	12	11	17	65					
				UC	IFX	Naïve	12	4	5	80					
				CD	IFX	Naïve	12	7	12	58					
155	Taxonera 2015 [166]	R	A	UC	IFX	Naïve					-	Response	54	79	68
		R	A	UC	IFX	Naïve					-	Remission	41	79	52
156	Taxonera 2014 [167]	R	A	CD	IFX	Naïve	13	16	59	27					
		R	A	UC	IFX	Naïve	9	16	38	42					
157	Taxonera 2017 (bis) [168]	R	A	UC	ADA	Naïve	24	12	68	18	RI or ID	Response	7	12	58
											RI or ID	Remission	2	12	17
					ADA	Non-naïve	24	64	116	55	RI or ID	Response	26	64	41
											RI or ID	Remission	13	64	20
158	Taxonera 2017 [169]	R	A	UC	GOL	Naïve and non-naïve	12	31	114	27	RI or ID	Response	22	31	71
159	Taxonera 2011 [170]	R	A	UC	ADA	Non-naïve	12	11	30	37	RI	Response	8	11	73
160	Tigue 2017 [171]	R	A	UC	IFX + ADA	-	12	3	38	8					
				CD	IFX + ADA		12	2	24	8					
161	Tkacz 2014 [172]	R	A	CD	IFX	Naïve	9	18	106	17					
162	Tursi 2018 [173]	R	A	UC	ADA	Naïve and non-naïve	18	9	56	16					
163	Vahabnezhad 2014 [174]	R	A+C	CD	IFX	Naïve	30	65	89	73	RI or ID	Response	40	65	62
				UC	IFX	Naïve	25	7	13	54	RI or ID	Response	4	7	57
164	Vanassche 2012 [175]	P	A	CD	IFX	Naïve	12	6	37	16					
165	Vandevondel 2018 [176]	R	A	UC	ADA	Naïve and non-naïve	6	129	231	56	RI	Response	77	129	60
166	Vatansever 2014 [177]	P	A	CD	IFX+ADA	-	12	3	35	9					
167	Verstock 2018 [178]	R	A	CD	ADA	Naïve	12	27	116	23	-	Response	19	27	70
				CD	ADA	Naïve	18	43	116	37					
168	Viazis 2015 [179]	P	A	CD	IFX+ADA	Naïve	28	31	132	23	RI or ID	Remission	25	31	81
169	Watanabe 2014. [180]	P	A	CD	ADA	Naïve and non-naïve	34	40	79	51	DI	Response	8	8	100
												Remission	6	8	75
170	West 2008 [181]	R	A	CD	ADA	No naïve	12	8	30	27	RI	Response	6	8	75
171	Wolf 2014 [182]	P	A	UC	ADA	Naïve and non-naïve	3	20	123	16	RI	Response	9	20	45
												Remission	4	20	20
172	Yamada 2014 [183]	R	A	UC	IFX	Naïve	36	17	24	71	RI or ID	Remission	16	17	94
173	Yokoyama 2016 [184]	R	A	CD	IFX+ADA	Naïve and non-naïve	18	8	107	7					

DI: Dose intensification. R: Retrospective. P: Prospective. UC: Ulcerative colitis. CD: Crohn’s disease. IFX: Infliximab. ADA: Adalimumab. CZP: Certolizumab pegol. GOL: Golimumab. *n*: number of patients undergoing dose intensification. N: total number of patients included. ID: Increase of dose. RI: Reduction of the interval of administration. n’: number of patients with a clinical response or remission after dose intensification. N’: total number of patients undergoing dose intensification.

**Table 2 jcm-10-02132-t002:** Dose intensification rate after the 12-month follow-up by the anti-TNF agent and medical condition.

Anti-TNF	UC/CD	DI Requirement (%, 95% CI)	*I2* (%)	Number of Included Studies
IFX	UC+CD	29 (22–36)	96	26
IFX	UC	40 (24–56)	97	8
IFX	CD	21 (15–28)	92	15
ADA	UC+CD	28 (22–34)	93	16
ADA	UC	29 (23–35)	86	6
ADA	CD	28 (17–38)	94	10

Anti-TNF: anti-tumor necrosis factor. UC: ulcerative colitis. CD: Crohn’s disease. DI: dose intensification. IFX: Infliximab. ADA: Adalimumab.

**Table 3 jcm-10-02132-t003:** The DI rate after 36-month follow-up by the anti-TNF agent and medical condition.

Anti-TNF	UC/CD	DI Requirement (%, 95% CI)	*I2* (%)	Number of Included Studies
IFX	UC+CD	38 (30–46)	96	15
IFX	UC	48 (34–62)	82	4
IFX	CD	35 (26–43)	96	12
ADA	UC+CD	24 (7–40)	92	4
ADA	UC	34 (3–64)	92	2
ADA	CD	3 (−4–11)	80	2

Anti-TNF: anti-tumor necrosis factor. UC: ulcerative colitis. CD: Crohn’s disease. DI: dose intensification. IFX: Infliximab. ADA: Adalimumab.

**Table 4 jcm-10-02132-t004:** Response rate by the anti-TNF agent and medical condition.

Anti-TNF	UC/CD	Response Rate (%, 95% CI)	*I2* (%)	Number of Included Studies
IFX	UC+CD	65 (49–80)	99	26
IFX	UC	62 (29–95)	99	8
IFX	CD	67 (59–75)	91	16
ADA	UC+CD	63 (55–70)	0	5
ADA	UC	58 (48–68)	NA	1
ADA	CD	69 (58–80)	0	4

Anti-TNF: anti-tumor necrosis factor. UC: ulcerative colitis. CD: Crohn’s disease. IFX: Infliximab. ADA: Adalimumab.

**Table 5 jcm-10-02132-t005:** Remission rate by the anti-TNF agent and medical condition in naïve patients.

Anti-TNF	UC/CD	Remission Rate (%, 95% CI)	*I2* (%)	Number of Included Studies
IFX	UC+CD	46 (34–59)	93	14
IFX	UC	50 (25–74)	96	7
IFX	CD	43 (33–53)	60	6
ADA	UC+CD	44 (31–58)	86	10
ADA	UC	17 (07–27)	NA	1
ADA	CD	50 (36–64)	79	8

Anti-TNF: anti-tumor necrosis factor. UC: ulcerative colitis. CD: Crohn’s disease. IFX: Infliximab. ADA: Adalimumab.

## Data Availability

Data sharing not applicable.

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
