# Peer review of "Frequency and Effectiveness of Empirical Anti-TNF Dose Intensification in Inflammatory Bowel Disease: Systematic Review with Meta-Analysis"

_jcm, 2021, doi:10.3390/jcm10102132_

Round 1

Reviewer 1 Report

The present work is well designed and rather exhaustive.

1. There are some inconsistencies that should be amended (e.g. page 1 line 34 and 39: inflammatory bowel disease versus IBD).

2. It is fundamental to add the pathophysiology of IBD in the introduction in order to demonstrate the pertinence of the use of anti-TNF in this disease and thus the relevance of the review.

4. In the “Literature search and study selection”, the authors refers to “only the most recent one was included”. Please, specify and justify the time interval.

5. In Table 1, the legend must come from above and the authors must repeat the headings on all pages.

6. In the discussion, authors have demonstrated the evidence existing in several papers already published on the topic. However, the Discussion section is quite poor: it is important to discuss about the data obtained. (e.g. page 6 line 222: Please report whether “DI was needed more often in patients with prior anti-TNF use.”

7. It is important to give possible future directions for this research.

Author Response

Dear Sir / Madam,

Enclosed you will find the revised paper in which the modifications suggested by the Reviewers have been made and are detailed as follows:

  1. There are some inconsistencies that should be amended (e.g. page 1 line 34 and 39: inflammatory bowel disease versusIBD).

    Thank you for your suggestion; it has been revised as requested; page 1 line 35.

  2. It is fundamental to add the pathophysiology of IBD in the introduction in order to demonstrate the pertinence of the use of anti-TNF in this disease and thus the relevance of the review.

We agree, thank you. Added as requested; page 1 lines 37-40.

  1. In the “Literature search and study selection”, the authors refers to “only the most recent one was included”. Please, specify and justify the time interval.

Thank you for the suggestion. This has been detailed as requested, page 2 line 67-70. When the literature search yielded two or more studies by the same author assessing the same population, only the most recent article was chosen, irrespective of the time interval, as it was assumed the latter published would include the most comprehensive and complete data.

  1. In Table 1, the legend must come from above and the authors must repeat the headings on all pages.

Thank you for your comment. Amended, as requested.

  1. In the discussion, authors have demonstrated the evidence existing in several papers already published on the topic. However, the Discussion section is quite poor: it is important to discuss about the data obtained. (e.g. page 6 line 222: Please report whether “DI was needed more often in patients with prior anti-TNF use.”

Thank you for your suggestion. More information about our conclusions regarding the results in both naïve and non-naïve patients has been added, as requested; page 17 lines 260-263.

  1. It is important to give possible future directions for this research.

Thank you, added as requested; page 18 lines 279-281; 302-303.

Yours faithfully,

Laura Guberna, Olga P. Nyssen, M. Chaparro and Javier P. Gisbert.

Reviewer 2 Report

This manuscript is well-written and well composed with a wide range literature search and comprehensive analyses.  Author have  presented some new findings through these researchs as follows; Rather high frequency of LOR in  the course of anti-TNF therapies. Dose intensification seem to be reasonable theprapy with good responses. However, there are many questions  how this situation should be improved. Early detection of LOR by therapeutic drug monitoring with proactive manner is one of the most reasonable answer to this question. But evaluation to this answer is not presented but a little discussed in this manuscript. Next step should be warranted on this arguement.

Author Response

Dear Sir / Madam,

Enclosed you will find the revised paper in which the modifications suggested by the Reviewers have been made.

Thank you very much for your kind response and your suggestion of considering proactive therapeutic drug monitoring as a way to improve early detection of loss of response. Precisely, in order to be able to assess empirical dose intensification, we have excluded those studies based on therapeutic drug monitoring so we cannot truly elaborate on this issue. However, there is undoubtedly a need to better characterize the risk factors influencing a loss of response and find potential strategies to prevent it.

Yours faithfully,

Laura Guberna, Olga P. Nyssen, M. Chaparro and Javier P. Gisbert.

Reviewer 3 Report

Thank you for the opportunity to revise the paper entitled “Frequency and effectiveness of empirical anti-TNF dose intesification in inflammatory bowel disease: systematic review with meta-analysis”.

The paper aims to quantify the overall rate of DI requirement for loss of response, to evaluate the efficacy and identify patients at risk for its requirement. A higher rate of DI requirement was observed in patients with prior exposure to anti-TNF agents compared to naïve group after 12 months follow up, however no difference in terms of efficacy occurred beetween groups. Furthermore, the need for DI was greater in UC than in patients with CD.

The topic is relevant because application of this therapeutic strategy is not well defined despite the extensive use in clinical practice. However, a few aspects need to be addressed.

1) Assessment of quality of randomized controlled trials and observational studies is lacking.

2) A subgroup analysis including RCTs should be performed.

3) How does this review differ from those already published on the same topic (ref. 185-188)? Are the results different? What did it add new to the literature? Please state this in the discussion.

4) I suggest adding in the conclusions section the need for a well stratified RCTs especially for UC because the rationale for DI in UC is not yet supported by RCTs.

Author Response

Dear Sir / Madam,

Enclosed you will find the revised paper in which the modifications suggested by the Reviewers have been made and are detailed as follows:

1) Assessment of quality of randomized controlled trials and observational studies is lacking.

Thank you for your comment. Initially, we discarded quality assessment due to the high number of studies (mainly observational) included in our study. This decision was taken post-hoc after performing an exploratory mapping review and confirming the wide range of observational studies in terms of number and design available in the literature responding to our topic of interest. We assumed this as a limitation of our study and this has been reported in the Discussion section, as appropriate.

However, thanks to your kind suggestion we reconsidered performing an evaluation of the RCTs only, as we had a decent sample size of patients included within this group and this could add robustness to our systematic review conclusions. And so, as per your request, the quality of the randomized controlled trials (RCTs) was assessed by means of the Cochrane Risk of Bias tool, and added as appropriate, page 17 line 233-240. Overall, most of the RCTs showed a low risk of bias in the majority of items assessed. Additionally, a subgroup-analysis was performed to control for heterogeneity in terms of study design, and no significant differences in DI requirement between RCTs and observational studies were reported.

2) A subgroup analysis including RCTs should be performed.

The subgroup analysis has been performed and the results have been included in the new heading “3.4. Randomized controlled trials” of the manuscript (page 17 line 220-224), as requested.

3) How does this review differ from those already published on the same topic (ref. 185-188)? Are the results different? What did it add new to the literature? Please state this in the discussion.

Thank you for your suggestion. When compared to previous literature reviews, our study includes a considerable higher number of studies and it assesses both ulcerative colitis and Crohn’s disease patients, which confers robustness and reliability to our work. This information has been included as requested; page 17 line 250-253.

4) I suggest adding in the conclusions section the need for a well stratified RCTs especially for UC because the rationale for DI in UC is not yet supported by RCTs

Thank you for your suggestion. It has been added as requested; page 18 line 279-281.

Yours faithfully,

Laura Guberna, Olga P. Nyssen, M. Chaparro and Javier P. Gisbert.